# Competition–Innovation Nexus: Product vs. Process, Does It Matter?

**Emil Palikot**

Stanford University, 450 Jane Stanford Way, Stanford, CA 94305, USA; palikot@stanford.edu

**Abstract:** I study the relationship between competition and innovation, focusing on the distinction between product and process innovations. By considering product innovation, I expand upon earlier research on the topic of the relationship between competition and innovation, which focused on process innovations. New products allow firms to differentiate themselves from one another. I demonstrate that the competition level that creates the most innovation incentive is higher for process innovation than product innovation. I also provide empirical evidence that supports these results. Using the community innovation survey, I first show that an inverted U-shape characterizes the relationship between competition and both process and product innovations. The optimal competition level for promoting innovation is higher for process innovation.

**Keywords:** market structure; innovation; competition

## 1. Introduction

Innovation has long been recognized as a key driver of economic growth and competitiveness, and much research has been dedicated to understanding the factors that encourage or hinder innovation (see, e.g., Solow (1956)). The role of market competition in promoting innovation has been a source of lively debate in economics for a few decades (Arrow 1962; Schumpeter 1982).

Aghion et al. (2005) propose an inverted U-shape hypothesis. They argue that two phenomena shape the impact of competition on innovation: (1) The escape competition effect, where firms innovate in order to move ahead of their rivals (fiercer competition creates larger innovation incentives), and (2) the Schumpeterian effect; this describes the innovation incentives of a firm using older technology that needs to be innovated to catch up with the technological frontier. The higher the post-innovation profit, the larger the incentive to innovate.

The theoretical model underpinning an inverted U-shape hypothesis considers technological progress as an improvement to the production cost (often referred to as process innovation). This paper extends this framework by considering product innovation. Product innovation captures the development of new or improved goods or services that offer better functionality or value to consumers. New products allow firms to differentiate themselves from the existing competition. In contrast, process innovation refers to the development of new or improved production or delivery methods. Process innovation is often associated with cost reduction, increased efficiency, and improved quality of goods and services. This distinction, commonly made in the literature, is important, as policies aimed at promoting innovation may need to be tailored to support either process or product innovation. This distinction is also empirically meaningful; for example, in the dataset that I analyze, more than one-third of firms that are involved in process innovations do not carry out product innovations.

In the model that I propose, the introduction of new products allows firms to differentiate themselves from one another. First, I demonstrate that the relationship between competition and innovation is characterized by an inverted U-shape in both process and

product innovations. Second, I compare the competition levels that maximize innovation incentivizes and show that they are higher for process innovations. The intuition behind this is that laggards lead to higher profits in markets where firms have different technologies (if the innovation is in the product rather than in the process). Product innovation results in differentiation, which means that the less-advanced firm also achieves some profit. This means that innovation incentives are smaller too.

I empirically test these results using data from the community innovation survey, a survey of over 90 thousand European firms. To proxy for innovation, I use innovation expenditures declared by the survey respondents; to measure competition, I consider the gross operating profit and turnover ratio. As innovation expenditures are highly dispersed, I use the negative binomial model. Finally, to address the endogeneity of competition and innovation, I propose an instrumental variable strategy, using foreign currency fluctuations as an instrument.

I provide empirical evidence supporting the hypotheses from the theoretical model. First, I demonstrate that there is an inverted U-shaped relationship, considering all innovation types. I then document them independently for product-focused sectors and process-oriented sectors. Finally, I demonstrate that the optimal competition level for creating innovation incentives is higher for sectors where technological progress mainly occurs through process innovation.

The rest of this paper is organized as follows. Section 2 presents a brief overview of the related literature. Section 3 presents a theoretical model. Section 4 presents a discussion on the empirical part of this paper. Section 5 concludes this paper.

## 2. The Competition–Innovation Debate

Joseph Schumpeter's provocative idea was that monopolistic profits, rather than fierce competition from rivals, are the main forces that motivate entrepreneurs to exploit new opportunities (Schumpeter 2013). Innovation changes industry structures. Technological progress is constantly creating monopolies by creating new markets and leaders. However, such monopolistic power is only temporary—it lasts until the next entrepreneur introduces an innovation to the market and replaces the incumbent. Joseph Schumpeter advocated for these types of monopolies as they yield high, temporary profits, resulting in powerful innovation incentives.

Investing in innovations is usually risky and almost always requires long-term financing. Consequently, financial markets might not allocate enough capital to such investments. For example, Gilbert (2006) argues that investors might be reluctant to engage in risky and long-term projects, as they might believe that a financially-strapped firm will ask for external financing for low-return projects, and rely on its internal resources for high-yield projects. In contrast, a monopolist can use obtained profits by exerting market power to finance long-term R&D.

Gilbert and Newbery (1982) show that a monopolistic market structure creates higher innovation incentives in the context of a patent race. Suppose that a company that spends the most on R&D obtains innovation with certainty; this resembles the auction of an exclusive good, where the agent with the highest valuation receives it. Consider a non-drastic innovation where a monopolist competes with a follower whose current technology does not yield a profit.[1] If the follower obtains the technology, it enters the market and competes with the monopolist. Thus, the follower compares the pre-innovation profits, which are zero, with the duopolistic profits. On the contrary, the monopolist compares the monopoly and duopoly profits. Therefore, the monopolist's innovation incentive reflects the profit stream that it will receive after foreclosing the market, which is likely higher than the duopoly profit that an entrant can achieve. This model has been further extended; Boone (2001); Salant (1984); Vickers (1985).

Dasgupta and Stiglitz (1980) consider the case where innovation is not exclusive; specifically, they develop a model where multiple firms can simultaneously develop similar innovations. They showcase a mechanism where several innovation incentives would

result in too-high R&D expenditures, increasing the overall welfare if only one firm were to invest in innovations. A monopolistic market structure prevents such overly high R&D investments.

In his seminal work from 1962, Kenneth Arrow challenges Joseph Schumpeter's results by showing another source of competition in innovation-centric markets (Arrow 1962). Consider a process innovation that decreases costs. A monopolist may not face competition today; however, if it innovates and strengthens its monopolistic position, some profit will be replaced. In other words, if the innovation succeeds, the monopolist will compare its current profits using old technology against higher profits. An innovator in a competitive market will compare its present low profits with possible higher ones from the new technology.

Consider a drastic innovation within a Bertrand competition framework. Symmetric firms using the same technology make no profit; however, if any one of those firms is able to decrease its costs, so much so that the new monopoly price is lower than the old marginal cost, then the innovating firm will go from having zero profit to a monopolistic profit with the new technology. On the other hand, a firm that has a monopolistic position using the old technology will have much lower net gains from the new drastic innovation. In the case of no market expansion, the profits from the new technology will simply replace the monopolistic profits from the old one. Thus, comparing the net gains in these two market structures, one can notice that there are more innovation incentives in a competitive organization.

The analysis of a non-drastic innovation case is more complicated because there is still pricing pressure from the other firms in a competitive market. Nevertheless, Arrow (1962) shows that, after accounting for the replacement effect, a monopolist will still have fewer innovation incentives. Greenstein and Ramey (1998) show the same mechanism in product innovation.

*Empirical Analyses*

Economic theories present various mechanisms—some highlight the higher innovation incentives in monopolistic structures, while others show that highly competitive markets lead to intense innovations. Which mechanism is dominating is ultimately an empirical question with the answer potentially depending on the characteristics of the sector.[2] MacDonald (1994) shows that relaxing import restrictions, which lead to greater competition from outside markets, resulted in an increase in labor productivity (US period 1972–1987). Cohen and Klepper (1996) study how spending on process innovation varies with respect to the firm size. They find a positive correlation between a firm's size and its spending on process R&D. Furthermore, Blundell et al. (1999) investigate the panel data of UK companies, and conclude that, on the one hand, these are high market share companies that commercialize the greater part of innovations. However, on the other, the increasing product market competition leads to more innovative activity. Lastly, they show that large companies benefit the most from innovations. Nickell (1996), using a panel of UK companies, shows that competition increases are associated with a total factor productivity growth. Several other papers investigating the competition–innovation debate find a positive relationship between the two; see Carlin et al. (2004); Okada (2005); Schmitz (2005).

The possibility of a non-linear relationship between competition and innovation is suggested in the meta-analysis by Kamien and Schwartz (1982). Scherer (1967) empirically documents the inverted U-shape in the relationship between R&D and market concentration. Aghion et al. (2001) further develop this line of reasoning by providing a theoretical foundation for such a result. In work by Aghion et al. (2001), two opposing mechanisms lead to an inverted U-shaped relationship: *the escape competition effect* captures innovation incentives in order to move away from a competitive market and gain monopolistic profits, while *the Schumpeterian effect* describes technological laggard incentives to innovate profitability and keeping up with the market. The theoretical model presented in this paper borrows heavily from this seminal work and extends it by considering both product and process innovations (unlike earlier works, which focused on process innovation).

Aghion et al. (2005), using panel data from UK manufacturing companies, provide supporting empirical evidence for an inverted U-shape hypothesis. The above-mentioned paper is a major step forward in a competition–innovation debate. This paper finds a strong quadratic relationship between competition and innovation using convincing econometrical methods, controlling for year and industry dummies, and addressing endogeneity with instrumental variables. Furthermore, one can compute the optimal competition amount in a particular sector by using this methodology. The inverted U-shaped hypothesis receives a great deal of attention and is examined in several other papers. For example, Tingvall and Poldahl (2006) test several competition measures and find a parabolic relationship with the Hirschman–Herfindahl index.

Several authors underline the importance of sector-specific effects; for instance, Hashmi (2013) and Cohen et al. (2000) suggest that the propensity to patent depends on the size of the firm, the sector where the company operates, and the type of innovation it focuses on. This paper contributes to this literature by considering the distinction between the type of innovation: product or process.

## 3. Theoretical Model

In this section, I extend the models of Aghion et al. (1997, 2001, and 2005) by introducing product differentiation, allowing firms to choose between product or process innovation. I first introduce the model and characterize the competition levels that maximize incentives to invest in product and process innovations. Finally, I derived the hypotheses that I test in Section 4.

*Fundamentals of the model.* An economy is populated with a unit mass of consumers who derive utility from the consumption of a continuum of goods. Each good comes from a different section; hence, there is a continuum of sectors. Sectors are duopolies. Firms manufacture using labor as the only input, based on a constant returns production function, and accept wages as a given. Thus, the unit's production cost is independent of the quantity produced. Each consumer supplies one unit of labor inelastically and has a consistent intertemporal discount rate $r$.

Consumer $i$ is characterized by a logarithmic instantaneous utility function:

$$U_i(x_{jt}) = \int_0^1 ln(\theta_{ij}s_{jk}x_j - p_{jk})dj, \tag{1}$$

where the utility of consumer $i$ from consumption in period $t$ is a sum of the consumption of goods from the $j$ sectors. All products available to consumers can be described in terms of $(s_{jk}, p_{jk})$, where $s_{jk}$ is the level of quality of a product of firm $k$ in sector $j$, and $p$ is the price of it.

Each consumer $i$ is characterized by a parameter $\theta_{ij}$ that measures his/her strength of preference of quality in sector $j$ and maximizes utility by choosing between firms $k$ and $k'$. Equation (2) describes the consumer's maximization problem in a sector:

$$\max_{x_k, x_{-k}} \theta_i s_k x_k + \theta_i s_{-k} x_{-k} \tag{2}$$

$$\text{subject to: } p_k x_k + p_{-k} x_{-k} \leq 1.$$

Assumption 1 describes the distributional assumption on the preference parameter $\theta$.

**Assumption 1.** *The taste parameter is (i) distributed uniformly on a unit interval between $\underline{\theta}$ and $\overline{\theta}$, i.e., $\theta \sim U(\underline{\theta}, \overline{\theta})$, (ii) $\overline{\theta} > 2\underline{\theta}$, and (iii) $c + \frac{\overline{\theta} - 2\underline{\theta}}{3}(s_k - s_{-k}) \leq \underline{\theta}s_k$.*

The first part of Assumption 1 is made to simplify solving the model. The second part ensures that there is sufficient heterogeneity in tastes across agents, and the third part guarantees that the market is covered. Together these assumptions ensure that the tastes are sufficiently differentiated so that firms might have the incentive to introduce new

products that differ in quality. However, this assumption is not innocuous and it restricts this analysis to the type of industries where products can be sufficiently differentiated.

*Taxonomy of sectors.* The technological level of a sector is characterized by two dimensions: The production efficiencies of the two companies, which are marginal costs $c_k$ and $c_{k'}$, and the quality of the two products, $s_k$ and $s_{k'}$. Firms engage in process innovations that decrease marginal costs or in product innovations that increase quality.

The company with better technology, lower marginal costs, or higher quality is referred to as a leader, and the other firm is referred to as a laggard. When both companies have technological advantages (e.g., one is more efficient and the other has a product of higher quality), the company that generates more profits will be called the leader. Thus, there are four types of sectors (see the Table 1):

**Table 1.** The taxonomy of sectors.

| Leveled sectors | Unleveled sectors in the process |
|---|---|
| $s_1 = s_2$ | $s_1 = s_2$ |
| $c_1 = c_2$ | $c_1 \neq c_2$ |
| Unleveled sectors in the product | Unleveled sectors in both dimensions |
| $s_1 \neq s_2 \ \Delta s = s_2 - s_1$ | $s_1 \neq s_2$ |
| $c_1 = c_2$ | $c_1 \neq c_2$ |

While I allow for firms to be differentiated in both dimensions, I describe cases where, at a specific period, firms engage in one of the innovation types. This simplification enables me to compare the relationship between competition and innovation in sectors dominated by product innovation with those where technological progress is driven by process improvements.

*Equilibrium profits of firms.* In each period $t$, firms set prices to maximize profits. Proposition 1 characterizes the equilibrium profits in each type of sector.

**Proposition 1.** *Equilibrium profits depend on the type of sector:*

- *In unleveled sectors of product quality, the profits are given by the following:*

$$\pi_k = \frac{\Delta s}{9}(2\bar{\theta} - \underline{\theta})^2 \text{ and } \pi_{k'} = \frac{\Delta s}{9}(\bar{\theta} - 2\underline{\theta})^2, \tag{3}$$

  *where k is the technological leader and k' is the laggard.*
- *In unleveled sectors of process technology, the profits are:*

$$\pi_k = c_{k'} - c_k \text{ and } \pi_{k'} = 0, \tag{4}$$

  *where k is the technological leader and k' is the laggard.*
- *In unleveled sectors in both dimensions, the profits are given by the following:*

$$\pi_k = \frac{1}{9}(\Delta s(2\bar{\theta} - \underline{\theta}) - c_k + c_{k'})^2 \text{ and } \pi_{-k} = \frac{1}{9}(\Delta s(\bar{\theta} - 2\underline{\theta}) + c_k - c_{k'})^2, \tag{5}$$

  *where k is the leader in the quality of the product, and k' is the firm with lower marginal costs.*
- *In leveled sectors, the profits of both firms are:*

$$\pi_k = \pi_{k'} = \alpha \times (\bar{\theta} - \underline{\theta}) \left[ \frac{\bar{\theta} - c}{2} \right]^2, \tag{6}$$

  *where $\alpha \in (0, 1/2)$ is a market conduct parameter, which takes the value of zero when firms are fully competing, and the value of $1/2$, when they are perfectly colluding, and c is the level*

*of marginal costs, I omit the subscript as both firms have the same level of marginal costs:*
$c = c_k = c_{k'}$.

**Proof.** See Appendix A for the proof of Proposition 1.　□

The difference in profits between the leader and laggard in sectors where firms differentiate in product quality depends on how differentiated the tastes of consumers are. Assumption 1 implies that $\bar{\theta} - 2\underline{\theta} < 2\bar{\theta} - \underline{\theta}$, which ensures that the profits of the leader are always higher than those of the laggard.

The proposed framework is asymmetric in the sense that the market conduct in leveled and unleveled sectors is modeled differently. I assume that firms in sectors with a clear leader and laggard consistently engage in competition. On the other hand, in leveled sectors, the model introduces the potential for collusive behavior, denoted by the $\alpha$ parameter. This theoretical construct is shaped by Assumption 2, which presumes a certain degree of collusion among firms. The main justification for this assumption is that homogeneous product characteristics can facilitate collusive behavior as the incentives to deviate from collusion are aligned (see work by Hay and Kelley (1974) for evidence supporting this).

**Assumption 2.** *I assume a minimum level of collusive behavior:*

$$
\alpha > \begin{cases} \frac{4\Delta s}{9} \frac{(\bar{\theta}-2\underline{\theta})^2}{(\bar{\theta}-\underline{\theta})(\bar{\theta}-c)^2} \\ \frac{4}{9} \frac{(\Delta s(\bar{\theta}-2\underline{\theta})+c_k-c_{-k})^2}{(\bar{\theta}-\underline{\theta})(\bar{\theta}-c)^2}, \end{cases} \tag{7}
$$

*where $\Delta$ measures the difference in technology between a leader and a laggard in an unleveled sector in product technology. I also assume that there is sufficient competition between firms, so that*

$$
\alpha < \begin{cases} \frac{c_{k'}-c_k}{\bar{\theta}-\underline{\theta}} \left[\frac{2}{\bar{\theta}-c}\right]^2 \\ \frac{4\Delta s}{9} \frac{(2\bar{\theta}-\underline{\theta})^2}{(\bar{\theta}-\underline{\theta})(\bar{\theta}-c)^2} \\ \frac{4}{9} \frac{\Delta s(2\bar{\theta}-\underline{\theta})-c_k-c_{k'}}{(\bar{\theta}-\underline{\theta})(\bar{\theta}-c)^2}. \end{cases} \tag{8}
$$

Assumption 2 ensures that profits in a leveled sector are lower than profits of the leader in an unleveled one but higher than those of the follower. Assumption 2 restricts the generality of the model to markets where unleveled firms are neither in perfect collusion nor in intense competition.

*Innovation Incentives*

The technological levels of companies are determined by their investment decisions. Following the work by Aghion et al. (1997), I assume that knowledge spillovers are high enough to guarantee that the highest permissible technological advantage is one step. As a consequence of the high knowledge spillover, the leader in an unleveled sector has no innovation incentives because its innovation would be automatically copied. This assumption makes it easier to solve the model analytically. Aghion et al. (2001) relax this assumption and numerically solve their model by showing that the presence of an inverted U-shape does not hinge on this assumption. This assumption has a different meaning in the product than in process innovation. While process innovation means that firms do not further decrease their marginal costs, product innovation implies that firms do not further differentiate themselves from one another.

Leveled companies and followers invest in R&D according to the cost function $\psi(n) = \frac{n^2}{2}$. Additionally, I assume that followers with probability $h$ might copy the innovation of the leader. Table 2 presents the innovation intensities of different players and the costs associated with them:

**Table 2.** Innovation intensities and associated costs.

| Type of Company | Innovation Intensity | Cost |
|---:|:---:|---:|
| leader | $0$ | $0$ |
| leveled firm | $n_0$ | $\frac{n_0^2}{2}$ |
| follower | $n_{-1} + h$ | $\frac{n_{-1}^2}{2}$ |

The equilibrium levels of innovation intensities $n_0$ and $n_{-1}$ are determined by necessary conditions for a symmetric Markov-stationary equilibrium, where each firm seeks to maximize the expected discounted profits. To solve them, I use a system of Bellman equations:

$$rV_1 = \pi_1 + (n_{-1} + h)(V_0 - V_1), \tag{9}$$

$$rV_{-1} = \pi_{-1} + (n_{-1} + h)(V_0 - V_{-1}) - \frac{n_{-1}^2}{2}, \tag{10}$$

$$rV_0 = \pi_0 + \bar{n}_0(V_{-1} - V_0) + n_0(V_1 - V_0) - \frac{n_0^2}{2}. \tag{11}$$

The current annuity value of being a leader $rV_1$ equals the flow of profits $\pi_1$ minus the expected loss of capital, which is the probability of successful innovation by the follower times the difference in profits if the innovation takes place. The value of being a follower comprises the profits in this period plus an expected capital gain, minus the costs of innovation. Being in a leveled sector, the worth is determined by the flow of profits, an anticipated capital loss if the rival company innovates (represented by $\bar{n}$ as the innovation intensity by the competing neck-and-neck company), an expected capital loss following a successful innovation, minus the cost of R&D.

Companies maximize their current value with respect to the innovation intensity; hence:

$$n_{-1} = V_0 - V_{-1}, \tag{12}$$
$$n_0 = V_1 - V_0. \tag{13}$$

in a Nash symmetric equilibrium $\bar{n}_0 = n_0$. Innovation intensities can be determined by solving the above system of Equation (9). Proposition 2 describes the solution.

**Proposition 2.** *The innovation intensity of leveled companies is:*

$$n_0 = -(r + h) + \sqrt{(r + h)^2 + 2\Delta\pi_1}, \tag{14}$$

*while the unleveled one is:*

$$n_{-1} = -(n_0 + r + h) + \sqrt{(n_0 + r + h)^2 + 2(\frac{n_0}{2} + (\pi_{-1} - \pi_0))}. \tag{15}$$

**Proof.**

$$r(V_1 - V_0) = \pi_1 - \pi_0 + (n_{-1} + h)(V_0 - V_1) - n_0(V_{-1} - V_0) + \frac{n_0^2}{2}$$

$$r(V_0 - V_{-1}) = \pi_0 - \pi_{-1} + n_0(V_{-1} - V_0) + n_0(V_1 - V_0) - \frac{n_0^2}{2} - (n_{-1} + h)(V_0 - V_{-1}) + \frac{n_{-1}^2}{2}$$

$$\frac{n_0^2}{2} + (r + h)n_0 - (\pi_1 - \pi_0) = 0$$

$$=> n_0 = -(r + h) + \sqrt{(r + h)^2 + 2\Delta\pi_1},$$

$$\text{where } \pi_0 = (1 - \Delta)\pi_1$$

$$\frac{n_{-1}^2}{2} + (r + n_0 + h)n_{-1} - \frac{n_0^2}{2} - (\pi_0 - \pi_{-1}) = 0$$

$$=> n_{-1} = -(n_0 + r + h) + \sqrt{(n_0 + r + h)^2 + 2(\frac{n_0}{2} + (\pi_{-1} - \pi_0))} \tag{16}$$

$\square$

By Proposition 2, the innovation intensity of the leveled sectors is an increasing function of the competition. Severe competition between firms in duopolistic markets reduces their profits and, therefore, their incentives to escape this state are high. The positive competition effect on innovation is the *escape competition effect*. The innovation intensity of followers is as follows: $n_{-1}$ is a decreasing function of competition. The innovation incentive of followers is shaped by the difference between the profits of a leveled company and a follower. Hence, if the profits of a leveled company are small because of the tight competition, incentives to reach this state are low. This effect is typically referred to as the Schumpeterian effect. These two effects are analogous to assumptions in work by Aghion et al. (1997, 2001).

The overall competition effect on innovation is a composition of these two forces: the *escape competition effect* and the *Schumpeterian effect*. Let us denote the steady-state probability of being in an unleveled sector as $\mu_1$ (and by $\mu_0$ of being in a leveled sector). The steady-state probability where, within one period, a sector moves from being an unleveled state to a leveled state is $\mu_1(n_{-1} + h)$, which is the probability of a sector being an unleveled state times the probability of a follower in that sector making a successful innovation. The probability of moving in the opposite direction is $2\mu_0 n_0$, which is the probability of a sector being leveled times the probability that any firm innovates. Proposition 3 describes the steady-state level of innovation intensity.

**Proposition 3.** *The flow of innovations is given by $I = \frac{4n_0(n_{-1}+h)}{n_{-1}+h+2n_0}$. Competition enters the equation through its influence on $n_0$. The relationship between competition and innovation intensity follows an inverted U-shape. The optimum competition level is a decreasing function of the profits of a laggard.*

**Proof.** In a steady state, the transition probabilities between leveled and unleveled states have to be equal:

$$\mu_1(n_{-1} + h) = 2\mu_0 n_0,$$
$$\mu_1 + \mu_0 = 1,$$
$$\mu_1 = \frac{2n_0}{n_{-1} + h + 2n_0}. \tag{17}$$

The flow of innovations $I$ is a sum of the innovation intensities coming from companies in both sectors:

$$I = 2\mu_0 n_0 + \mu_{-1}(n_{-1} + h) = 2\mu_1(n_{-1} + h) = \frac{4n_0(n_{-1} + h)}{n_{-1} + h + 2n_0}. \tag{18}$$

The innovation intensity of a leveled firm $n_0$ is an increasing function of competition. Hence, I proxy competition with $n_0$. □

Proposition 3 follows the same logic as in Aghion et al. (2005). Let $I^{process}$ be the innovation flow in the process-oriented industries and $I^{product}$ in product-oriented industries. Also, let $n^*_{0,process} = \arg\max I^{process}$ be the competition level that maximizes the flow of innovation in process-oriented industries and analogously $n^*_{0,product} = \arg\max I^{product}$. Proposition 4 compares the two.

**Proposition 4.** *The competition level, which maximizes innovation intensity, is higher in process-oriented sectors than in product-oriented sectors; $n^*_{0,process} > n^*_{0,product}$.*

**Proof.** This follows from the comparison of profit differences between unleveled sectors in process technology and leveled sectors with the differences in profits between unleveled sectors in product technology and leveled sectors described in Proposition 1. □

The lower optimal levels for product-oriented sectors can be interpreted as follows; the follower in an unleveled sector in product technology obtains higher profits than in a process sector because of a higher degree of differentiation between the companies. Thus, the difference between being a follower and a neck-and-neck company is smaller for the product-oriented sector. Therefore, the incentive to move from an unleveled state to a leveled state is also smaller.

## 4. Testing Theoretical Hypotheses

The theoretical discussion shows that the competition–innovation interplay depends on several characteristics of a sector, such as the competition mode and the technological spill-over level. The most obvious way to verify this hypothesis is to include sector dummy variables and check their significance. However, the mere fact that the relationship between competition and innovation varies from sector to sector is of limited use. The focus here is to determine meaningful rules that would provide further insights into the competition–innovation nexus. In some sectors, firms might focus on process innovations that offer a competitive edge in the form of lower marginal costs and, others might focus on developing new products, which may result in the creation of new markets or major changes in demand in existing markets. This section aims to empirically test the hypotheses emerging from the theoretical model.

*Hypotheses from the theoretical model.* I am interested in testing three hypotheses, based on results from the theoretical model.

**Hypothesis 1.** *The relationship between competition and innovation in process-oriented sectors follows an inverted U-shape.*

**Hypothesis 2.** *The relationship between competition and innovation in product-oriented sectors follows an inverted U-shape.*

**Hypothesis 3.** *The competition level that maximizes the flow of innovation is higher in process-oriented sectors than in product-oriented sectors.*

### 4.1. Discussion of the Dataset and the Main Variables

Measuring both competition and innovation is difficult. Firstly, endogeneity in the relationship is likely. Firms that successfully innovate might grow faster and influence the sector's structure. Secondly, the data are often of poor quality. The most commonly used measures of innovation are patents and R&D expenditures, both of which have severe drawbacks. Patents might fail to describe the actual innovativeness of a sector. Some companies might decide not to file for patent protection; this might be particularly true in countries where intellectual property rights are not successfully executed or product innovation is more likely to be patented when compared to process innovation because enforcement of a process innovation patent may be difficult, and applying for it makes it public (Levin et al. 1987). R&D expenditures are often not reported or are misreported. Furthermore, expenditures on research do not capture the quality. Table A1 in Appendix B summarizes the most commonly used indicators.

I use an alternative metric to capture the innovation activities of firms—the community innovation survey (CIS). The CIS is a survey of innovation activities of enterprises, conducted by Eurostat. I use data from the 2006–2008 wave. The harmonized survey is designed to provide information on the innovativeness of sectors by the enterprise type, the different types of innovation (i.e., newly introduced products or process innovations), and various aspects of the development of an innovation, such as the objectives, information sources, public funding, innovation expenditures, etc. The CIS provides statistics broken down by country, innovator types, economic activities, and size classes. In total, data encompass 90,274 companies from 15 European countries.

To construct a measure of innovativeness, I use companies that indicate that they are innovators, based on their positive responses to one (or more) of the following questions from the CIS questionnaire:

1. *During the years 2006–2008, did your enterprise introduce new or significantly improved goods? (exclude the simple resale of new goods purchased from other enterprises and changes of a solely aesthetic nature.)*
2. *During the years 2006–2008, did your enterprise introduce new or significantly improved services?*
3. *During the years 2006–2008, did your enterprise introduce new or significantly improved methods of manufacturing or producing goods or services?*
4. *During the years 2006–2008, did your enterprise introduce new or significantly improved logistics, delivery, or distribution methods for inputs, goods, or services?*
5. *During the years 2006–2008, did your enterprise introduce new or significantly improved support activities for your processes, such as maintenance systems or operations for purchasing, accounting, or computing?*

I classify a company as an innovator if it provided a positive response to any of the questions above. The intensity of innovative activities is quantified by multiplying a binary variable (indicating if an enterprise is an innovator) by the sum expended on innovations by that particular enterprise in 2008. Questions 1 and 2, which focus on new goods or services, pertain to product innovation. Conversely, questions 3, 4, and 5, which inquire about production methods, logistics and distribution, and other supporting activities, are associated with process innovation. In the empirical analysis, a firm is considered a product innovator if it responds positively to either the first or second question, or both. Similarly, a firm is labeled a process innovator if it answers affirmatively to at least one of questions three through five.

The community innovation survey's main strength lies in its expansive coverage of firms. However, as a survey, this dataset is not without its limitations. Specifically, in this context, firms self-report whether they introduced product or process innovation. This approach might encounter issues, such as potential recall inaccuracies or misunderstandings about the distinction between various categories by the individuals completing the survey. Furthermore, there might be discrepancies in how different firms interpret these categories. Finally, I do not observe the exact amount spent on the product or process innovation, just the total amount. Thus, in empirical analyses where product and process innovations are treated separately, firms that invest in both innovation types have inflated amounts of total investments.

To measure the competition amount, I construct the following variable:

$$Competition_i = \frac{1}{\sum_{i \in I} \frac{\pi_i}{r_i}},$$ (19)

where $I$ is an industry, firms are mapped to one sector in my dataset, $\pi_i$ is the gross operating profit, which is the difference between revenue and the cost of making a product or providing a service, not accounting for overhead, payroll costs, and before taxes, and interest, and finally, $r_i$ is the turnover of firm $i$.

Data come from the Structural Business Statistics provided by Eurostat; these are data from surveys conducted in all European Union member states at the company level. My metric is very similar to the Lerner index used by Aghion et al. (2005); Nickell (1996); the difference is that the Lerner index accounts for differences in financial costs. In my data, I do not have access to financial costs per sector.[3]

The summary statistics are presented in Table 3. One can notice that innovation intensity is highly skewed as there are some firms with high innovation expenditures included in my sample.

**Table 3.** Summary statistics. Innovation intensity in thousands of dollars.

| Statistic | N | Mean | St. Dev. | Min | Pctl (25) | Pctl (75) | Max |
|---|---|---|---|---|---|---|---|
| Innovation intensity | 64,509 | 579.565 | 20,956.400 | 0 | 0 | 3.3 | 3,800,000 |
| Competition | 60,054 | 0.119 | 0.100 | 0.000 | 0.072 | 0.145 | 1.364 |
| Competition squared | 60,054 | 0.025 | 0.124 | 0.0004 | 0.005 | 0.021 | 1.860 |

### 4.2. Econometric Model

By construction, my innovation measure has many zeroes (firms that do not invest in innovations) and is always positive. Therefore, as the main specification, I consider the Poisson model (as a robustness check, I also include a Negative Binomial model to account for over-dispersion). The aim is to estimate the following regression:

$$\text{Innovation Intensity}_{it} = \beta_0 + \beta_1 \text{competition}_{jt} + \beta_2 \text{competition}_{jt}^2 + X'\gamma + \epsilon_{jt}, \quad (20)$$

$X$ is a vector of control variables, including country dummies, to control for country-specific effects. It also includes the size of the company in 2008, aggregated into three groups: small enterprises (below 50 employees); medium (between 50 and 250 employees); and large ones. It also accounts for the firms' revenue and the change in revenue. In all estimations, I use robust standard errors.

The relationship between competition and innovation follows an inverted U-curve if coefficients of Equation (20) have the following signs: $\beta_1 > 0$ and $\beta_2 < 0$. This pattern enables the calculation of the optimal competition level for promoting innovation:

$$\frac{\partial \text{Innovation Intensity}}{\partial Competition} = \beta_1 + 2\beta_2 Competition^* = 0,$$

$$\Rightarrow Competition^* = -\frac{\beta_1}{2\beta_2}. \quad (21)$$

Innovation intensity is measured at a firm level, and in many cases, firms invest in both process and product innovations. This could be the case, for example, when a new product is introduced together with a new process. Consequently, many sectors will be engaged in both innovation types. The optimal competition level for supporting, for example, process innovation in this sector might not be optimal for product innovation. The lack of a clear-cut distinction between sectors only characterized by product innovation and those only characterized by process is a limitation in my approach.[4]

The relationship between competition and innovation is likely endogenous. The competitive pressure posed by other market participants influences a firm's innovation incentives. However, after a successful innovation, the company gains a competitive advantage over its rivals; hence, the organization of the market will be influenced. To address this concern, I adopt the strategy proposed in Revenga (1992).

Respondents to the community innovation survey questionnaire were asked whether the focus of their operations was international (inside and outside of the European Union). Around 18 thousand enterprises provided affirmative responses to this question. The appreciation of the currency of that producer increases the competitive pressure it faces. Products of rivals become more competitive as their prices drop and the producer needs to respond to keep its customers. Data on currency fluctuations come from Eurostat (real effective exchange rate (deflator: consumer price indices—18 trading partners—Euro area)).

The nonlinearity of the model poses restrictions on the choice of the instrumenting strategy. I use the control function approach; here, this boils down to a two-step procedure. I first regress currency fluctuations on my competition measure. Second, I include residuals from the first step in the final estimation. In the nonlinear model, control function estimates are not the same as 2SLS estimates using any choice of instrument. In control, the function is likely to be more efficient but less robust. For a further discussion, see Blundell and Powell (2003).

### 4.3. An Inverted U-Shaped Hypothesis

I start by re-evaluating an inverted U-shaped hypothesis in my empirical context. Table 4 shows the results.

**Table 4.** Inverted U-shapes of all innovation types.

| | Dependent Variable: | | |
|---|---|---|---|
| | Innovation Intensity | | |
| | **(1)** | **(2)** | **(3)** |
| Competition | 7.914 *** (2.941) | 6.039 ** (2.832) | 10.220 *** (3.845) |
| Competition Squared | −7.671 ** (3.390) | −5.377 ** (2.317) | −9.267 * (5.131) |
| Country FE | Yes | Yes | Yes |
| Size, turnover, growth | No | Yes | Yes |
| Sample | All firms | All firms | Only exporting firms |
| Observations | 60,054 | 56,632 | 12,049 |

Note: * $p < 0.1$; ** $p < 0.05$; *** $p < 0.01$. All innovation types. Poisson models. In the first model, only country-fixed effects are controls; in the second model, there are additional control variables: turnover in 2008—an indicator of whether it is a large firm or not, and an indicator of growth in turnover; the last model—the subset of exporting firms and the control function.

Statistically significant inverted U-shapes appear in all specifications. I conclude that there is a negative quadratic relationship between competition and innovation.[5]

Product vs. Process Innovation

The aim of this section is to provide an empirical test for hypotheses H1, H2, and H3. The distinction between optimal competition levels for promoting product or process innovation is evaluated by using the product or process innovation as a dependent variable. Table 5 presents the results.

In both baseline specifications, the entire sample with no control function, I estimate an inverted U-shaped relationship for both process and product innovations. Specifications three and six, which focus on the sub-sample of exporting firms and include the control function, have lower statistical power, and the coefficient on the quadratic term is statistically insignificant. Nevertheless, the point estimates show inverted U-shapes. I find that the optimal competition level is higher for process innovation: 0.47 vs. 0.53 (for models in columns 1 and 4) and 0.53 to 0.58 (columns 2 and 5). Thus, one cannot reject the H3.

Finally, in Table 6, I account for the over-dispersion of the outcome variable. First, I trim the data on the 95th percentile and then use the negative binomial model. I present the specification that focuses on the exporter's subsample and has the control function.

**Table 5.** Inverted U-shape: process vs. product innovation.

| | *Dependent Variable:* | | | | | |
|---|---|---|---|---|---|---|
| | Process Intensity | | | Product Intensity | | |
| | **(1)** | **(2)** | **(3)** | **(4)** | **(5)** | **(6)** |
| Competition | 8.097 *** (2.982) | 6.471 ** (2.929) | 10.645 ** (4.420) | 10.202 *** (2.982) | 8.110 *** (2.929) | 14.973 ** (7.054) |
| Competition Squared | −7.706 ** (3.485) | −5.603 ** (2.409) | −9.954 (7.923) | −10.797 *** (3.485) | −7.337 *** (2.409) | −22.805 (19.728) |
| Country FE | Yes | Yes | Yes | Yes | Yes | Yes |
| Size, turnover, growth | No | Yes | Yes | No | Yes | Yes |
| Sample | All firms | All firms | Exporting firms | All firms | All firms | Exporting firms |
| Observations | 60,054 | 56,632 | 12,049 | 60,054 | 56,632 | 12,049 |

Note: ** $p < 0.05$; *** $p < 0.01$. The first three models are for process innovation; the last three models are for product innovation. Poisson models. The first and fourth models are only for country-fixed effects as controls; the second and the fifth models are additional control variables: turnover in 2008—this is an indicator of whether it is a large firm or not, as well as the growth in turnover; the third and the last model—the subset of exporting firms and the control function.

**Table 6.** Product vs. process—negative binomial model.

| | *Dependent Variable:* | |
|---|---|---|
| | **Process Intensity** (1) | **Product Intensity** (2) |
| Competition | 1.681 *** (0.631) | 2.020 *** (0.624) |
| Competition Squared | −1.329 *** (0.473) | −1.868 *** (0.591) |
| Country FE | Yes | Yes |
| Size, turnover, growth | Yes | Yes |
| Sample | Exporting firms | Exporting firms |
| Observations | 11,446 | 11,443 |

Note: *** $p < 0.01$. Negative binomial models with a control function on the subsample of exporting firms. The first model considers process intensity as the dependent variable; the second model considers product intensity.

Estimates presented in Table 6 also show an inverted U-shaped relationship between competition and innovation for both process and product innovations. The optimal competition level is higher in process innovation (0.63) than in product innovation (0.54).

Overall, I cannot reject H1, H2, or H3. I provide empirical evidence that is consistent with the predictions of the theoretical model. The relationship between competition and innovation follows inverted U-shapes for both process and product innovations. The competition level that maximizes innovation appears to be higher in process innovations.

## 5. Conclusions

The interplay between competition and innovation has been a source of controversy and a lively economic debate for at least several decades (Arrow 1962; Schumpeter 1982). More recent papers have introduced the hypothesis that the relationship between competition and innovation is characterized by an inverted U-shape; these papers provide empirical evidence to support this (Aghion et al. 1997, 2005). The logic underlying this line of research is that the net competition effect on innovation is an interplay between two contradicting forces. The escape competition effect, where firms innovate in order to escape fierce competition and start reaping monopoly profits, accounts for the positive impact. In contrast, the Schumpeterian effect, where laggards innovate so that they can achieve a level of technology that allows them to compete in the market, accounts for the negative part of the relationship. The net competition effect on innovation depends on the relative strengths of these effects, which are endogenized in the model.

In this paper, I extend this reasoning to a setting where there are two innovation types: process and product. Process innovation is modeled as cost improvements, while product innovation involves the introduction of new products, leading to product differentiation. First, in a theoretical model, I demonstrate that an inverted U-shaped relationship emerges in both cases. However, the competition level that maximizes the intensity of innovation likely differs between the two innovation types. In particular, I demonstrate that the optimal competition level is higher for process innovations.

Second, I empirically test inverted U-shaped hypotheses using the community innovation survey (CIS). The CIS dataset allows me to differentiate between sectors that introduce process innovations and those where technological progress occurs through product innovation. I start by showing that an inverted U-shaped relationship holds when considering any type of innovation. Next, I separately analyze the competition's impact on process and product innovations. I find inverted U-shapes in both cases. Furthermore, consistent with the theoretical model, I find that the optimal competition level for promoting innovation is higher in process innovation.

My empirical strategy addresses the issue of endogeneity between competition and innovation by focusing on the subset of firms that are exporters and using currency fluctuations as an instrument. I also propose a negative binomial model, which accounts for the over-dispersion of innovation metrics.

The paper has several limitations. First, the proposed theoretical model introduces a somewhat asymmetric treatment of leveled and unleveled sectors. Specifically, the model posits that firms with equivalent technological capacities engage in partial collusion while firms with disparate technological capacities consistently compete. Furthermore, the model assumes that the extent of firm collusion falls within a defined interval, wherein the minimum and maximum competition levels increase with the technological disparity, both in terms of quality and costs, between the leading and lagging firms. This assumption plays a significant role in the model because it implies that leveled firms do not compete away all their profits but rather share a surplus. Consequently, this assumption amplifies the innovation incentives with the goal of attaining technological parity while it dampens the innovation incentives with the intent of escaping competition. This assumption limits the range of industries or markets to which the model is applicable.

Second, the primary variables deployed in this research encompass firm investments in product and process innovations derived from the community innovation survey data. The differentiation between product and process innovations is based on the survey questionnaire, a method that carries certain limitations. Specifically, my approach assumes that firms can clearly distinguish between innovations centered on the development and introduction of new products or services and those targeted at enhancing processes. Furthermore, it assumes that firms treat these innovation types as distinct entities. This approach, however, would not be applicable in scenarios where the development of every new product is accompanied by a new process or in circumstances where process innovation is rare and predominantly occurs alongside the introduction of new products. In such contexts, the delineation between product and process innovations becomes blurry and potentially meaningless. Additionally, as in any self-reported survey data, inaccuracies due to recall, exaggeration, or desirability bias are possible. Moreover, in the CIS data, I do not observe the specific amounts spent on product and process innovations, only the total expenditure and an indicator of whether the firm has been involved in product or process innovation. In practice, firms might be involved in both innovation types, spreading their R&D budgets between the two types. The lack of this specific distinction introduces an important mismeasurement in the data used for this analysis.

Given these constraints, future research could potentially refine the model to incorporate a more nuanced understanding of competition dynamics, as well as leverage more granular data to differentiate between investments in product and process innovations. More comprehensive datasets could offer deeper insights into how firms allocate their R&D budgets across different innovation types. Additionally, future studies could consider how other factors, such as firm size, industry characteristics, or regulatory environments, interact with competition to influence innovation.

In conclusion, the findings of this paper underscore the significance of differentiating between product and process innovations when examining the competition–innovation nexus. As a policy implication, fostering the appropriate competition level, based on whether the focus is on product or process innovation, could stimulate greater innovation within various industry sectors. These results underscore that innovation policies ought to be fine-tuned to specific industries and market circumstances.

**Funding:** This research received no external funding.

**Data Availability Statement:** Restrictions apply to the availability of these data. Data was obtained from Eurostat and are available upon request and approval from Eurostat. Details of the procedure to access microdata of Community Innovation Survey are here: https://ec.europa.eu/eurostat/web/microdata/community-innovation-survey.

**Acknowledgments:** I would like to thank Marc Ivaldi, the participants of the seminars, the Toulouse School of Economics, and the two anonymous referees. This paper is based on data from the Eurostat, community innovation survey 2006–2008. The conclusions drawn from the data lie entirely with the author.

**Conflicts of Interest:** The author declares no conflicts of interest.

**Appendix A. Proof of Proposition 1**

**Proof.** Unleveled sectors in product quality: without a loss of generality, I am assuming that $s_k > s_{-k}$. Given the assumption that consumers purchase only a unit of each good, and their preferences are distributed randomly between $\underline{\theta}$ and $\bar{\theta}$, I can determine an indifference condition: $\hat{\theta}s_k - p_k = \hat{\theta}s_{-k} - p_{-k}$. Consumers with $\theta_i < \hat{\theta}$ will consume goods of inferior quality, and those with higher quality will purchase a superior product. Therefore, there is demand directed at both companies:

$$D_{k'} = \frac{p_k - p_{k'}}{\Delta s} - \underline{\theta}$$
$$D_k = \bar{\theta} - \frac{p_k - p_{k'}}{\Delta s},$$

where $\Delta$ measures the difference between the technologies of the leader and the laggard. Thus, companies solve the maximization problem:

$$\max_{p_k}(p_k - c_k)D_k(p_k, p_{k'}) \tag{A1}$$

which gives the best response functions:

$$p_{k'} = \frac{p_k - \underline{\theta}\Delta s + c}{2}$$
$$p_k = \frac{\bar{\theta}\Delta s + p_{k'} + c}{1}$$

Equilibrium prices:

$$p_{k'} = \frac{(\bar{\theta} - 2\underline{\theta})\Delta s}{3} + c$$
$$p_k = \frac{(2\bar{\theta} - \underline{\theta})\Delta s}{3} + c$$

Equilibrium profits:

$$\pi_{k'} = \frac{\Delta s}{9}(\bar{\theta} - 2\underline{\theta})^2 \tag{A2}$$

$$\pi_k = \frac{\Delta s}{9}(2\bar{\theta} - \underline{\theta})^2 \tag{A3}$$

The assumption on the distribution of $\theta$: $\bar{\theta} - 2\underline{\theta} < 2\bar{\theta} - \underline{\theta}$ guarantees that profits of the leader are higher than those of the laggard.

Unleveled industries in process technology: In unleveled industries in process technology, the quality of the goods is the same; therefore, consumers are indifferent between purchasing one good or the other as long as the price is the same. However, companies differ in production efficiency. This setup results in a very fierce competition mode. Again, I assume that company $k$ is the more efficient one, $c_k < c_{k'}$.

Consumer problem:

$$\max_{x_k, x_{k'}} \theta_i s x_k + \theta_i s x_{k'}$$
$$\text{subject to: } p_k x_k + p_{-k} x_{k'} \leq 1 \tag{A4}$$

By a standard Bertrand undercutting argument, there are two equilibrium candidates:

$$\begin{cases} p_k = c_{k'} \\ p_{k'} = c_{k'} + \epsilon \end{cases} \quad \begin{cases} p_k = c_{k'} - \epsilon \\ p_{k'} = c_{k'} \end{cases}$$

To focus on the interesting case, where the production is profitable, I assume that $\theta s - c_{k'} \geq 0 \; \forall \theta$. Profits are given by:

$$\pi_1 = c_{-1} - c_1 \tag{A5}$$

$$\pi_{-1} = 0 \tag{A6}$$

*Unleveled sectors on both dimensions.* As before I am assuming that company $k$ is the leader in the quality of the product. Here, I also assume that company $k'$ is the more efficient one. Hence, companies solve the profit maximization problem:

$$\max_{p_k}(p_k - c_k)[\bar{\theta} - \frac{p_k - p_{k'}}{\Delta s}] \tag{A7}$$

$$\max_{p_{k'}}(p_{-k} - c_{k'})[\frac{p_k - p_{k'}}{\Delta s} - \underline{\theta}] \tag{A8}$$

The solution to this problem gives the best response function and prices:

$$p_k = \frac{\bar{\theta}\Delta s + p_{k'} + c_k}{2}$$

$$p_{k'} = \frac{p_k + c_{k'} - \underline{\theta}\Delta s}{2}$$

$$p_k^* = \frac{\Delta s(2\bar{\theta} - \underline{\theta}) + 2c_k + c_{k'}}{3} \tag{A9}$$

$$p_{k'}^* = \frac{\Delta s(\bar{\theta} - 2\underline{\theta}) + c_k + 2c_{k'}}{3} \tag{A10}$$

and the profits are described by:

$$\pi_k = \frac{1}{9}(\Delta s(2\bar{\theta} - \underline{\theta}) - c_k + c_{k'})^2 \tag{A11}$$

$$\pi_{k'} = \frac{1}{9}(\Delta s(\bar{\theta} - 2\underline{\theta}) + c_k - c_{k'})^2 \tag{A12}$$

*Leveled sectors.* In a leveled sector, companies use the same technology to produce their goods, which are considered identical by consumers. If companies engage in a fierce price competition, they will end up making zero profits. The organization of the sector, a duopoly, facilitates collusive behavior. Therefore, the profits of the companies will be proportional to the amount of collusion. Firms will capture the $\alpha$ share of the monopolistic profit, where $\alpha \in (0, 1/2)$, which is

$$\max_{p}(p - c)(1 - F(p)), \text{ where}$$

$$F(p) = \frac{p - \underline{\theta}}{\bar{\theta} - \underline{\theta}}$$

$$\pi^M = (\bar{\theta} - \underline{\theta})[\frac{\bar{\theta} - c}{2}]^2$$

$$\pi_0 = \alpha\pi^M$$

$\square$

## Appendix B. Commonly Used Metrics of Innovation

**Table A1.** The most commonly used indicators of competition and innovation. Source: Gilbert (2006) and my own research.

| Author | Measures of R&D | Measure of Competition |
|---|---|---|
| Scherer (1965) | Patents, R&D Employment | Firm size, market concentration |
| Scherer (1967) | R&D Employment | Market concentration |
| Comanor (1967) | R&D Expenditures | Market concentration |
| Mansfield (1968) | R&D Expenditures | Firm size |
| Mansfield et al. (1977) | R&D Expenditures, innovations | Firm size, market concentration |
| Link (1980) | Rate of Return on R&D | Firm size |
| Mansfield (1981) | R&D Expenditures | Firm size, market concentration |
| Scherer (1983) | R&D Expenditures, patents | Firm size |
| Link and Lunn (1984) | Rate of Return on R&D | Market concentration |
| Levin and Reiss (1984) | R&D Expenditures | Market concentration |
| Hall et al. (1984) | R&D Expenditures | Firm size |
| Scott (1984) | R&D Expenditures | Firm size, market concentration |
| Culbertson and Mueller (1985) | R&D employment, expenditures, patents | Firm size, market concentration |
| Levin et al. (1985) | R&D Expenditures, innovations | Market concentration |
| Angelmar (1985) | R&D Expenditures | Market concentration |
| Lunn and Martin (1986) | R&D Expenditures | Firm size, market concentration |
| Lunn (1986) | Patents | Market concentration |
| Acs and Audretsch (1987) | Number of innovations | Firm size |
| Nickell (1996) | Total factor productivity | Number of competitors |
| Blundell et al. (1999) | Number of innovations, patents | Market share |
| Aghion et al. (2005) | R&D Expenditures, patents | Lerner index |
| Hashmi (2013) | Patents | Lerner index |

## Appendix C. Firms with Non-Zero Innovation Expenditures

The decision for a firm to pursue innovation could potentially be independent of the amount of resources allocated to such endeavors. If that is true, the proposed theory explains the intensity of innovation, rather than the decision of whether to become an innovator or not. To address this distinction, this section focuses on a specific subset of firms, identified as innovators, which have dedicated non-zero amounts to innovation. I further explore the influence of competition on the level of their innovation expenditure. I carry out this analysis separately for process and product innovators and compare the optimal competition level. The findings are presented in Table A2.

I find that an inverted U-shaped relationship holds in the subsample restricted to firms that are innovators. Furthermore, comparing the coefficients associated with product and process innovations, I find that the optimal competition level to maximize the innovation expenditure amount is higher in the process-oriented sector rather than in the product oriented sector.

**Table A2.** Competition–innovation relationship for the subset of firms that have non-zero innovation expenditures.

| | *Process Intensity* | | | *Product Intensity* | | |
|---|---|---|---|---|---|---|
| | **(1)** | **(2)** | **(3)** | **(1)** | **(2)** | **(3)** |
| Competition | 7.728 *** (2.908) | 5.895 ** (2.877) | 10.133 *** (3.894) | 9.144 *** (2.908) | 7.596 *** (2.877) | 15.136 *** (3.894) |
| Competition squared | −7.006 ** (3.092) | −5.068 ** (2.301) | −9.073 * (5.400) | −9.176 *** (3.092) | −6.588 *** (2.301) | −23.082 *** (5.400) |
| Country FE | Yes | Yes | Yes | Yes | Yes | Yes |
| Size, turnover, growth | No | Yes | Yes | No | Yes | Yes |
| Sample | All firms | All firms | Exporting firms | All firms | All firms | Exporting firms |
| Observations | 56,306 | 53,070 | 11,299 | 53,443 | 50,339 | 10,978 |

Note: * $p < 0.1$; ** $p < 0.05$; *** $p < 0.01$. The first three models are for process innovations; the last three models are for product innovation. Poisson models. The first and fourth models are only country-fixed effects as controls; the second and fifth models are additional control variables: turnover in 2008 is an indicator of whether it is a large firm or not, as well as growth in turnover; the third and last model—the subset of exporting firms and the control function. The sample used in the first three models only includes firms that have positive expenditures on process innovation and the latter three firms that have positive expenditures on product innovation.

## Notes

[1]　A drastic innovation is the type of innovation that reduces marginal costs to a degree where the monopoly price is lower than the competitive price, with old technology Arrow (1962). Analogously, drastic product innovation will offer quality improvement, to the extent that competitors are driven out of the market, even if they price at cost.

[2]　See Kamien and Schwartz (1982) for a thorough review of the early empirical literature on the topic).

[3]　An important limitation of the competition metric used in this paper, as well as the Lerner index, is that it ignores the market structure. That is, different combinations of individual firms' profits can lead to the same outcome.

[4]　As an additional robustness check, in Appendix C, I focus on the subsample, which only includes firms that spent non-zero amounts on innovations. The results that I obtain are qualitatively the same.

[5]　To include standard diagnostic checks, I consider a linear model and estimate it using the 2SLS estimator. In a sample of exporters, I reject the hypothesis that the instruments are weak with a *p*-value that is less than 0.001 and the Hausman test statistic is 16.82.

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
