# Peer review of "Competition–Innovation Nexus: Product vs. Process, Does It Matter?"

_econometrics, doi:10.3390/econometrics11030021_

Round 1

Reviewer 1 Report

General Comments:

The paper presents an analysis of the relationship between competition and innovation, specifically focusing on process and product-oriented sectors. The theoretical framework is well-developed, and the empirical analysis using the Community Innovation Survey (CIS) dataset provides valuable insights. Overall, the paper contributes to the existing literature on the topic. However, there are some areas that could be further improved to strengthen the study.

Specific Comments and Suggestions:

1. Clarify the Assumptions: The paper relies on several assumptions regarding collusive behavior, knowledge spillovers, and competition. It would be helpful to provide more explicit justifications and explanations for these assumptions. Additionally, consider discussing the potential limitations or caveats of these assumptions.

2. Data Limitations: The paper acknowledges the limitations of using patents and R&D expenditures as measures of innovation. However, it would be beneficial to discuss the limitations of the CIS data as well, such as potential biases in self-reported innovation activities. Consider addressing any concerns regarding the reliability and validity of the CIS data and discuss how they may impact the findings.

3. Robustness Checks: While the paper includes some robustness checks, it would be valuable to conduct additional tests to further strengthen the findings. For example, consider alternative econometric models or additional control variables to assess the robustness of the inverted U-shape relationship.

4. Distinction between Product and Process Innovations: Clarify the differentiation between product and process innovations in the CIS dataset. Discuss the potential limitations of relying on self-reported survey responses and how this may affect the interpretation of the results.

5. Implications and Policy Relevance: In the conclusion section, provide a discussion on the practical implications of the findings and their relevance for policymakers. How can the results inform strategies to promote innovation in both process and product-oriented sectors? Consider discussing potential policy recommendations or avenues for future research in this area. 

Other minor comments and suggestions. 

1. Please move all commas after Equations (3)-(7) to the ends of the corresponding equations. Check the entire manuscript for this kind of issue. 

2. Please specify the meaning of "+" in the denominator in Equation (15).  

3. If an equation is at the end of a sentence, please put a period after the equation. Check the entire manuscript for this kind of issue. 

Overall, the paper presents a comprehensive analysis of the relationship between competition and innovation in process and product-oriented sectors. Addressing the suggested improvements would further strengthen the study and its contribution to the field.

Author Response

Dear Reviewer,

Many thanks for your comments. I appreciate the time and effort spent reading and commenting on this manuscript. I agree with all your remarks and I believe that I addressed all the points that you raised. Below, I paste your comments and my replies (in italics).

Specific Comments and Suggestions:

  1. Clarify the Assumptions: The paper relies on several assumptions regarding collusive behavior, knowledge spillovers, and competition. It would be helpful to provide more explicit justifications and explanations for these assumptions. Additionally, consider discussing the potential limitations or caveats of these assumptions.

I expanded parts of section 3 where these assumptions are made by adding statements that the assumptions restrict the generality of the model. I also provided more intuition for these assumptions. Additionally, I expanded paragraphs in the conclusion describing this.

  1. Data Limitations: The paper acknowledges the limitations of using patents and R&D expenditures as measures of innovation. However, it would be beneficial to discuss the limitations of the CIS data as well, such as potential biases in self-reported innovation activities. Consider addressing any concerns regarding the reliability and validity of the CIS data and discuss how they may impact the findings.

I added a paragraph in the conclusion and a discussion in section 4 describing the limitations of the CIS data.

  1. Robustness Checks: While the paper includes some robustness checks, it would be valuable to conduct additional tests to further strengthen the findings. For example, consider alternative econometric models or additional control variables to assess the robustness of the inverted U-shape relationship.

I added the Appendix C where I focus on the subset of firms that have non-zero innovation expenditure. This achieves two objectives, first I was worried that the main results are driven by the high number of zeroes and second, it’s plausible that the decision of whether to be an innovator or not, is different from the decision on the amount of innovation expenditure, and the theory that I propose focuses on the intensity of innovation. I found that my results hold in this subsample.

  1. Distinction between Product and Process Innovations: Clarify the differentiation between product and process innovations in the CIS dataset. Discuss the potential limitations of relying on self-reported survey responses and how this may affect the interpretation of the results.

I added these definitions on page 14. Just below them, I mentioned the potential limitations of using the CIS. These are also reiterated later in the conclusion.

  1. Implications and Policy Relevance: In the conclusion section, provide a discussion on the practical implications of the findings and their relevance for policymakers. How can the results inform strategies to promote innovation in both process and product-oriented sectors? Consider discussing potential policy recommendations or avenues for future research in this area. 

Added two paragraphs about this in the conclusion

Other minor comments and suggestions. 

  1. Please move all commas after Equations (3)-(7) to the ends of the corresponding equations. Check the entire manuscript for this kind of issue. 

Done

  1. Please specify the meaning of "+" in the denominator in Equation (15).  

That was a typo, now it’s fixed

  1. If an equation is at the end of a sentence, please put a period after the equation. Check the entire manuscript for this kind of issue. 

Done

Again I am grateful for your comments.

Best wishes,

Author Response

Dear Reviewer,

Many thanks for your comments. I appreciate the time and effort spent reading and commenting on this manuscript. I agree with all your remarks and I believe that I addressed all the points that you raised. Below, I paste your comments and my replies (in italics).

Major Remarks

  • In page 7, Equation (6), the letter ? is not introduced.

Thanks for pointing this out. I added the definition. In the leveled sectors firms have the same level of costs, hence I dropped the subscript. Now this is explained.

  • In page 10, Equation (15), the + is not explained.

This was a typo, now it’s fixed.

Minor Remarks

  • In page 6, under Assumption 1, there is a typo "... and the third path ..." should be "... and the third part ..."

Fixed. I also checked the rest of the documents for this type of issues.

  • In page 8, Equation (8), in the middle term, it is better use "4Δ?"

Changed

Thank you for your feedback. I hope you will be satisfied with the changes.

Best wishes,

Round 2

Reviewer 2 Report

Thank you for your great effort in clarifying and correction. It has good shape now.